# Health care workers in conflict and post-conflict settings: Systematic mapping of the evidence

**Lama Bou-Karroum**[1,2], **Amena El-Harakeh**[1,3], **Inas Kassamany**[2], **Hussein Ismail**[4], **Nour El Arnaout**[5], **Rana Charide**[4], **Farah Madi**[4], **Sarah Jamali**[6], **Tim Martineau**[7], **Fadi El-Jardali**[1,2,8], **Elie A. Akl**[1,3,8,9] *

**1** Center for Systematic Reviews on Health Policy and Systems Research (SPARK), American University of Beirut, Beirut, Lebanon, **2** Department of Health Management and Policy, Faculty of Health Sciences, American University of Beirut, Beirut, Lebanon, **3** Clinical Research Institute (CRI), American University of Beirut Medical Center, Beirut, Lebanon, **4** Faculty of Health Sciences, American University of Beirut, Beirut, Lebanon, **5** Global Health Institute, American University of Beirut, Beirut, Lebanon, **6** Faculty of Medicine, American University of Beirut, Beirut, Lebanon, **7** Department of International Public Health, Liverpool School of Tropical Medicine, United Kingdom, **8** Department of Health Research Methods, Evidence, and Impact (HEI), McMaster University, Hamilton, Ontario, Canada, **9** Department of Internal Medicine, Faculty of Medicine, American University of Beirut, Beirut, Lebanon

* ea32@aub.edu.lb

**Data Availability Statement:** All relevant data are within the paper and its Supporting Information files.

## Abstract

### Background

Health care workers (HCWs) are essential for the delivery of health care services in conflict areas and in rebuilding health systems post-conflict.

### Objective

The aim of this study was to systematically identify and map the published evidence on HCWs in conflict and post-conflict settings. Our ultimate aim is to inform researchers and funders on research gap on this subject and support relevant stakeholders by providing them with a comprehensive resource of evidence about HCWs in conflict and post-conflict settings on a global scale.

### Methods

We conducted a systematic mapping of the literature. We included a wide range of study designs, addressing any type of personnel providing health services in either conflict or post-conflict settings. We conducted a descriptive analysis of the general characteristics of the included papers and built two interactive systematic maps organized by country, study design and theme.

### Results

Out of 13,863 identified citations, we included a total of 474 studies: 304 on conflict settings, 149 on post-conflict settings, and 21 on both conflict and post-conflict settings. For conflict settings, the most studied counties were Iraq (15%), Syria (15%), Israel (10%), and the

**Funding:** EAA and FEJ received funding from the Lebanese National Council for Scientific Research (CNRS)-American University of Beirut (AUB) and the Alliance for Health Policy and Systems Research of the World Health Organization (WHO). The funders had no role in study design, data collection and analysis, decision to publish, or preparation of the manuscript.

**Competing interests:** The authors have declared that no competing interests exist.

State of Palestine (9%). The most common types of publication were opinion pieces in conflict settings (39%), and primary studies (33%) in post-conflict settings. In addition, most of the first and corresponding authors were affiliated with countries different from the country focus of the paper. Violence against health workers was the most tackled theme of papers reporting on conflict settings, while workforce performance was the most addressed theme by papers reporting on post-conflict settings. The majority of papers in both conflict and post-conflict settings did not report funding sources (81% and 53%) or conflicts of interest of authors (73% and 62%), and around half of primary studies did not report on ethical approvals (45% and 41%).

## Conclusions

This systematic mapping provides a comprehensive database of evidence about HCWs in conflict and post-conflict settings on a global scale that is often needed to inform policies and strategies on effective workforce planning and management and in reducing emigration. It can also be used to identify evidence for policy-relevant questions, knowledge gaps to direct future primary research, and knowledge clusters.

## Introduction

Health care workers (HCWs) are essential for the delivery of health care services in conflict areas and in rebuilding health systems post-conflict. However, HCWs in conflict areas around the world are being threatened, detained, and killed. For instance, in Syria, Physicians for Humans Rights has reported that, since the start of the conflict till December 2017, 847 medical personnel have been killed [1]. In Afghanistan, around 92 attacks against health facilities and health workers killed 14 health workers and four caretakers in the period extending from March 1, 2015 till February 10, 2016 [2].

Direct attacks and insecurity have led to the exodus of HCWs from conflict areas. In Syria, 50% of the health workers and 95% of physicians living in Aleppo have left the country since 2011. In Iraq, almost half of the health professionals have emigrated since 2014 [2]. In Nigeria, almost all health workers have escaped areas controlled by Boko Haram since 2012, leading to the closure of 450 health facilities [2].

The resulting shortage of HCWs has devastating effects on the delivery of, and access to health care not only during conflicts but also in the aftermath of war. The post-conflict settings are characterized by poor health outcomes due to limited availability of HCWs and disruption of health systems [3, 4]. Rebuilding the health workforce is critical to address health needs and strengthen health systems. Furthermore, post-conflict settings present a window of opportunity to develop responsive and evidence-informed strategies and policies to address defects in the supply, distribution and performance of the health workforce [4, 5].

In 2012, the World Health Organization (WHO) passed a resolution that calls on the WHO Director General for leadership in documenting evidence of attacks against health workers, facilities, and patients in situations of armed conflict [6]. A scoping review on HCWs in Syria and other "Arab Spring" countries showed scarcity of research evidence on HCWs in the setting of the "Arab Spring" [7]. While that review revealed a number of themes of interest (e.g., violence against health care workers, education, practicing in conflict setting, migration), it focused on only one region and did not address post-conflict settings. Therefore, the objective

of this study was to systematically identify and map the published evidence on HCWs in both conflict and post-conflict settings. Our ultimate aim is to inform researchers and funders on research gap on this subject and support relevant stakeholders by providing them with a comprehensive resource of evidence about HCWs in conflict and post-conflict settings on a global scale.

## Methods

### Study design

This systematic mapping was based on a protocol registered with Open Science Framework [8]. We followed standard methodology for screening, data extraction and coding, data analysis, and visualizing the findings in systematic mapping. Contrary to systematic reviews, systematic mapping does not aim to answer a specific question but instead "collates, describes and catalogues available evidence (e.g. primary, secondary, quantitative or qualitative) relating to a topic of interest" [9]. In accordance with the definition of systematic maps, this study is a "systematic visual presentation of the availability of relevant evidence," but not the content of the evidence [10] for the topic of health care workers in conflict and post-conflict settings. The studies included in a systematic map can be used to identify evidence for policy-relevant questions, knowledge gaps to direct future primary research, and knowledge clusters. Knowledge clusters are sub-sets of evidence that may be suitable for secondary research, for example systematic review.

### Eligibility criteria

**Population of interest.**   Our population of interest consisted of any personnel providing health services such as: midwives, nurses, paramedics, pharmacists, physicians, laboratory technicians, community health workers as well as medical students and trainees. We excluded military HCWs, because we aimed to focus on the delivery of health care primarily to civilians.

**Setting of interest.**   We included both conflict and post-conflict settings. We considered both conflicts between and within states [11]. We focused on contemporary conflicts that started after or were ongoing in the 1990s. We defined conflicts as international armed conflicts between two or more states or non-international armed conflicts between non-governmental armed groups or with governmental forces [11]. Post-conflict settings are considered as a stage of recovery of the state from a conflict or crisis and a stage of rebuilding and reconstruction starting from emergency and stabilization followed by transition and recovery, and peace and development [5, 12, 13]. We referred to the description used by the authors when specifying whether the setting of the study was conflict or post-conflict.

**Study design.**   We included all types of study designs, including news, editorials, commentaries, opinion pieces, technical reports, primary studies, narrative reviews, and systematic reviews. We excluded conference abstracts. We restricted our eligibility criteria to papers published after the year 2000 to better reflect the current challenges facing health systems and the new aspects of contemporary conflicts.

### Literature search

We searched the following electronic databases: Medline (Ovid), PubMed, EMBASE (Ovid), the Cochrane Central Register of Controlled Trials (CENTRAL), Cumulative Index of Nursing and Allied Health Literature CINAHL (EBSCOT) on July 2017. We also searched the ReBUILD Consortium Resources webpage and the Human Resources for Health (HRH) Global Resource Center.

We used both index terms and free text words for the two following concepts: (1) health care workers and (2) conflict and post-conflict settings. The search terms and Medical Subject Headings (MeSH) terms for each database were developed with the guidance of an information specialist. We did not limit the search to specific languages. S1 File provides the search strategies for the different databases.

## Selection process

**Title and abstract screening.** Teams of two reviewers used the above eligibility criteria to screen titles and abstracts of identified citations in duplicate and independently for potential eligibility. We retrieved the full texts for citations judged as potentially eligible by at least one of the two reviewers.

**Full-text screening.** Teams of two reviewers used the above eligibility criteria to screen the full texts in duplicate and independently for eligibility. The teams of two reviewers resolved disagreement by discussion or with the help of a third reviewer. We used standardized and pilot-tested screening forms. We conducted calibration exercises to ensure the validity of the selection process.

## Data extraction and coding

Two reviewers extracted data using standardized and pilot tested forms. The reviewers resolved any disagreement by discussion and when needed with the help of a third reviewer. We conducted calibration exercises to ensure the validity of the data abstraction process.

We extracted from each paper the following information:

- Citation;

- Year of publication;

- Countr(ies) subject of the paper;

- Type of publication (e.g., news, editorial, correspondence, opinion pieces, primary study, narrative review, systematic review, case study, technical report)

- Language of publication;

- Authors' information:

  ○ Total number of authors;

  ○ Number of authors from the countr(ies) subject of the paper;

  ○ Country of affiliation of the first author;

  ○ Country of affiliation of the contact author;

- Characteristics of the journal of publication (name and impact factor);

- Setting (conflict or post-conflict);

- Theme(s) of the study for conflict settings: we adopted the themes from a previous scoping review on health care workers in the setting of Arab Spring [7]: violence against health care workers, education, practicing in conflict setting, migration, and other (S2 File);

- Theme(s) of the study for post-conflict settings: we adopted the theme(s) from a previous review on human resource management in post-conflict health systems [5]: workforce supply, workforce distribution, workforce performance, and other (S2 File);

- Reporting of funding of the study;

- Reporting of conflict of interest of authors;

- Ethical approval of the study.

For the themes, data was coded as 'other' if it did not address any of the existing themes, or if it covered an emerging theme and an existing one. Using an iterative process of review and refinement, data coded as 'other' was revisited, collated and new themes were generated.

### Critical appraisal

We did not appraise the quality of included studies since our review is consistent with standard systematic mapping methodology [9].

### Data analysis

We conducted a descriptive analysis of the general characteristics of the included papers using frequencies. We also used the results of this review to build two interactive and visual systematic evidence maps on (1) HCWs in conflict settings and (2) HCWs in post-conflict settings. We represented the evidence maps by country, type of publication and themes. We have also provided direct links to the included studies in the maps.

## Results

### Study selection

Fig 1 summarizes the study selection process. Out of 13,863 identified unique citations, we included a total of 474 studies [4, 14–470]: 304 on conflict settings, 149 on post-conflict settings, and 21 on both conflict and post-conflict settings. We excluded 968 papers for the following reasons: not study design of interest (n = 63); not setting of interest (n = 324); not population of interest (n = 538); and not timeframe of interest (e.g. ceasefire was called on before 1990) (n = 43). We present below our findings on the characteristics of the included papers, journals, authors, funding, conflicts of interest, and ethics reporting. We also report on the two generated systematic maps.

### Characteristics of the included papers

Fig 2 presents a geographical map of the countries focus of the included papers (S3 File). A description of the included studies is also presented in S4 File. The majority of the articles included in this systematic mapping were about low-income countries. For conflict settings (N = 325), 79% of the included papers addressed specific conflicts related to 47 countries. The most studied counties were Iraq (15%), Syria (15%), Israel (10%), and the State of Palestine (9%). For post-conflict settings (N = 170), 91% of the included papers addressed specific settings related to 32 countries. Sierra Leone (14%) was the most studied country followed by Uganda (11%) and Afghanistan (9%). The majority of papers on conflict and post-conflict were published in English language (98% and 100% respectively).

Table 1 represents the themes focus of the papers on health care workers in conflict (N = 325) and post-conflict (N = 170) settings. More than one theme was reported in 33% of the papers on HCWs in conflict settings and in 52% of the papers on HCWs in post-conflict settings. In addition to the themes about HCWs in conflict settings reported in Bou-Karroum et al. (2018) [7], three additional themes emerged in this review, and those were the role of HCWs in peace promotion or protecting health care, mental health of HCWs, and medical ethics. Most of the included papers on conflict settings addressed the theme of violence against

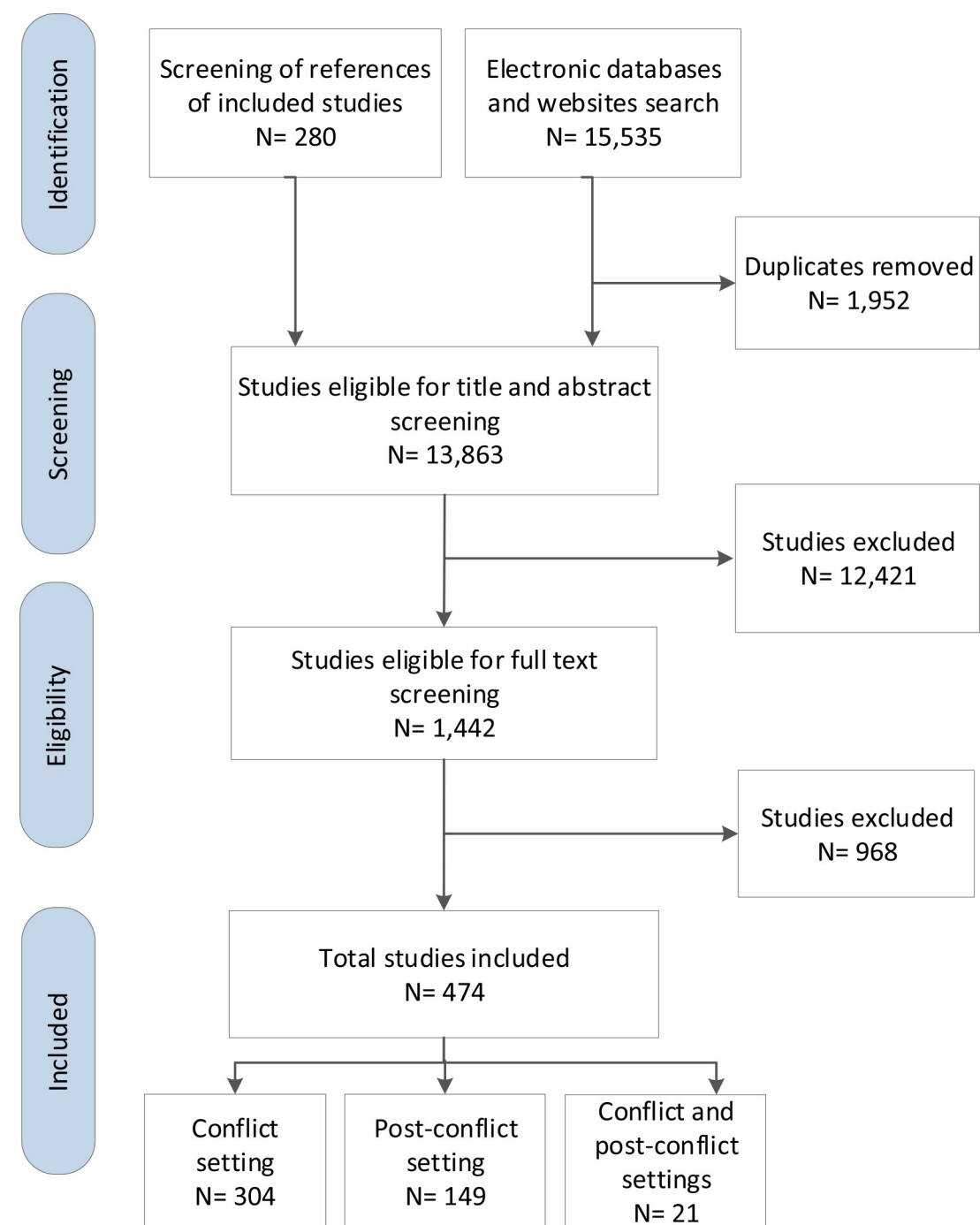

**Fig 1. Preferred Reporting Items for Systematic reviews and Meta-Analyses (PRISMA) study flow diagram for selection.**

health workers (41%), followed by health or medical practice (34%) and education (21%). For post-conflict settings, besides the themes adopted from Roome et al. (2014) [5], an emerging theme was the mental health of HCWs. The majority of the included papers on post-conflict settings addressed the workforce performance theme (77%) followed by workforce supply (58%).

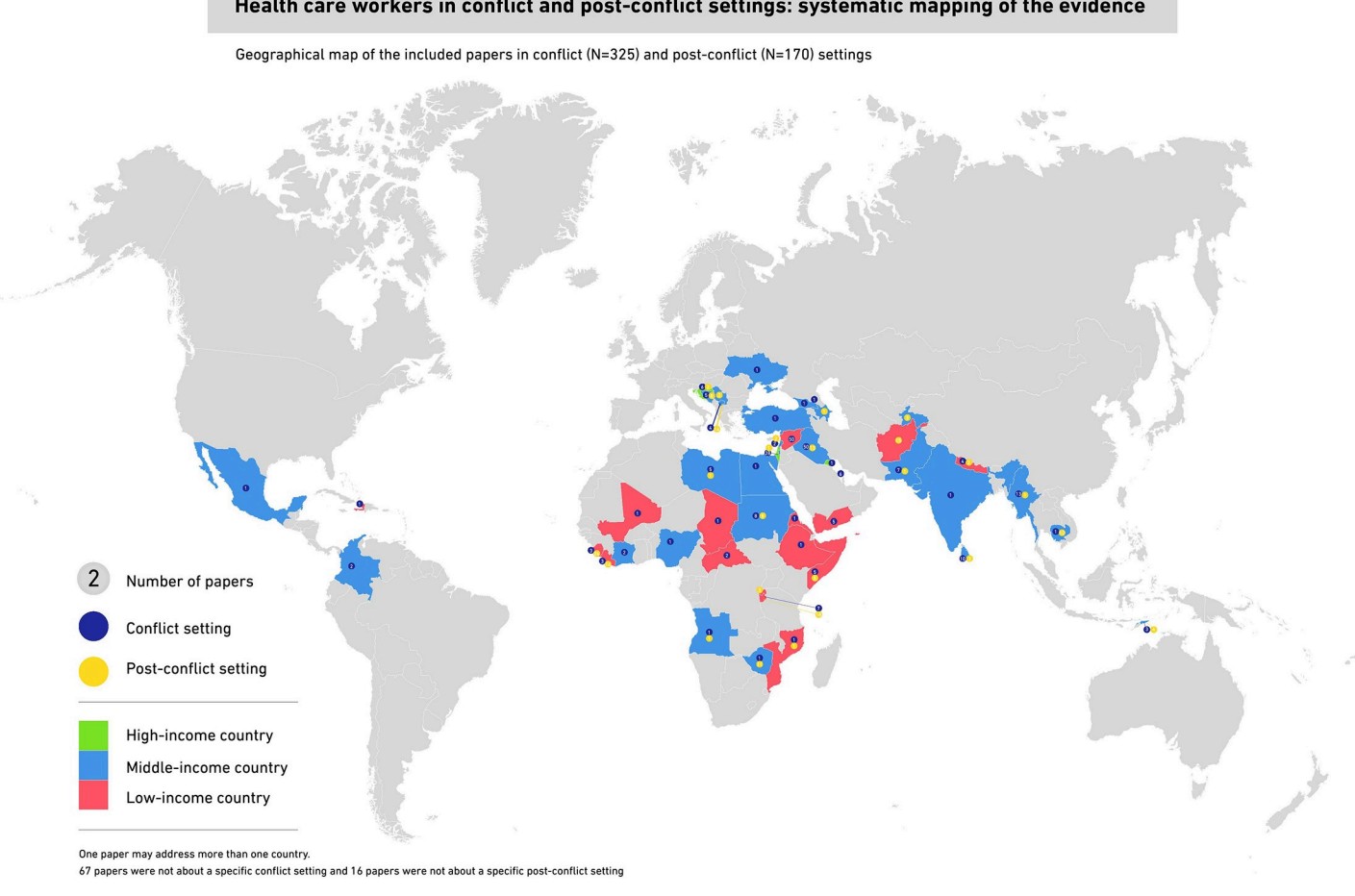

**Fig 2. Geographical map of the included papers in conflict (N = 325) and post-conflict (N = 170) settings.**

Fig 3 shows the annual production rate of the included papers. The year for the peak number of publications was 2013 for conflict settings and 2014 for post-conflict settings. Fig 4 shows the types of publication of the included papers. Opinion pieces represented the most common type of publication (39%) in conflict settings, followed by primary studies (23%) and news (18%). Primary studies were the most common type of publication (33%) in post-conflict settings followed by technical reports (24%) and case studies (21%).

## Characteristics of the journals

For conflict settings (N = 325), the included papers were published across 134 journals. The journals that published the highest proportions of included studies were the Lancet (15%), BMJ (7%), and CMAJ (5%). Out of the 134 journals, 90 journals (67%) had 2017 impact factors. 230 of the 325 papers were published in these 90 journals and had a median impact factor of 4.74 (IQR = 1.74–27.94).

For post-conflict settings (N = 170), the included papers were published across 79 journals. The journals that published the highest number of included studies were the Lancet (6%), Conflict and Health (5%), and Health Policy and Planning (5%). Out of the 79 journals, 51 journals (65%) had 2017 impact factors. 92 of the 170 papers were published in these 51 journals and had a median impact factor of 2.42 (IQR = 1.61–3.31).

**Table 1. Topics focus of the papers on health care workers in conflict (N = 325) and post-conflict (N = 170) settings.**

| Topics of the papers* | n (%) |
|---|---|
| **Conflict setting (N = 325)** | |
| • Violence against health care workers | 133 (41) |
| • Health or medical practice | 109 (34) |
| • Education | 67 (21) |
| • Role in peace promotion or protecting health care | 48 (15) |
| • Mental health | 42 (13) |
| • Migration | 37 (11) |
| • Medical ethics | 16 (5) |
| **Post-conflict setting (N = 170)** | |
| • Workforce performance | 131 (77) |
| • Workforce supply | 98 (58) |
| • Workforce distribution | 40 (24) |
| • Retention | 22 (13) |
| • Mental health | 8 (5) |

*One paper may address more than one topic.

## Characteristics of the authors

Table 2 summarizes the characteristics of the authors of the included papers. For conflict settings, 69% of the included papers reported affiliations of authors and addressed specific conflict(s). Out of these, 40% had at least 1 author affiliated with the country focus of the paper. The median percentage of authors affiliated with the country focus of the paper was null (0%) (IQR = 0–75). In addition, most of the first and corresponding authors were affiliated with countries different from the country focus of the paper (68% and 70% respectively), mainly the United States of America (42% and 39% respectively) followed by the United Kingdom (21% and 24% respectively).

For post-conflict settings, 75% of the included papers reported affiliations of authors and addressed specific conflict(s). Out of these, about half (53%) had at least 1 author affiliated with the country focus of the paper. The median percentage of authors affiliated with the country focus of the paper was 20% (IQR = 0–67). Similar to conflict settings, most of the first and corresponding authors were affiliated with countries different from the country focus of the paper (66% and 75% respectively), mainly the United Kingdom (37% and 40% respectively) followed by the United States of America (32 and 31% respectively).

## Funding, conflicts of interest and ethics reporting characteristics

Table 3 shows the funding, conflicts of interest, and ethics reporting characteristics of the included papers. For conflict settings, most of the included papers did not report funding sources (81%) or statements of conflicts of interest of authors (73%). Out of the included primary studies, about half (55%) reported ethical approval to conduct the studies. For post-conflict settings, about half of the included papers did not report funding sources (53%) and 62% did not report on the conflicts of interest of authors. Out of the included primary studies, 59% reported ethical approval.

## Systematic maps

The two systematic maps, which represent a visual and interactive overview of the evidence on health care workers in conflict and post-conflict settings, can be freely accessed and

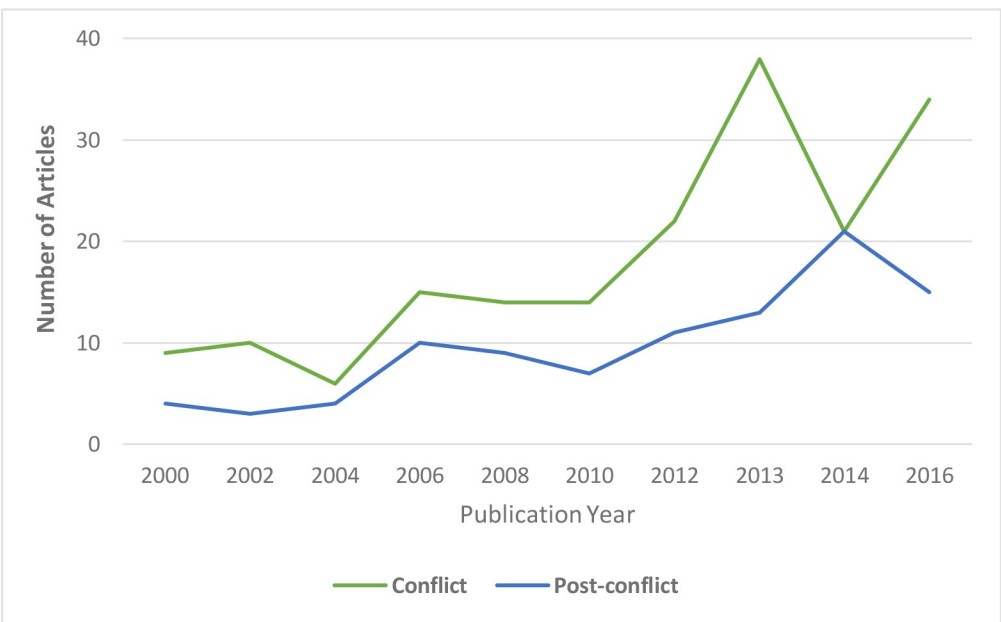

**Fig 3. Publication year of articles of the papers included in the systematic maps in conflict (N = 325) and post-conflict settings**.

downloaded using the following links for conflict settings (http://evidencemaphcw.com/gapmap/conflict) and for post-conflict settings (http://evidencemaphcw.com/gapmap/post-conflict). The maps allow data to be filtered and sorted by type of primary studies

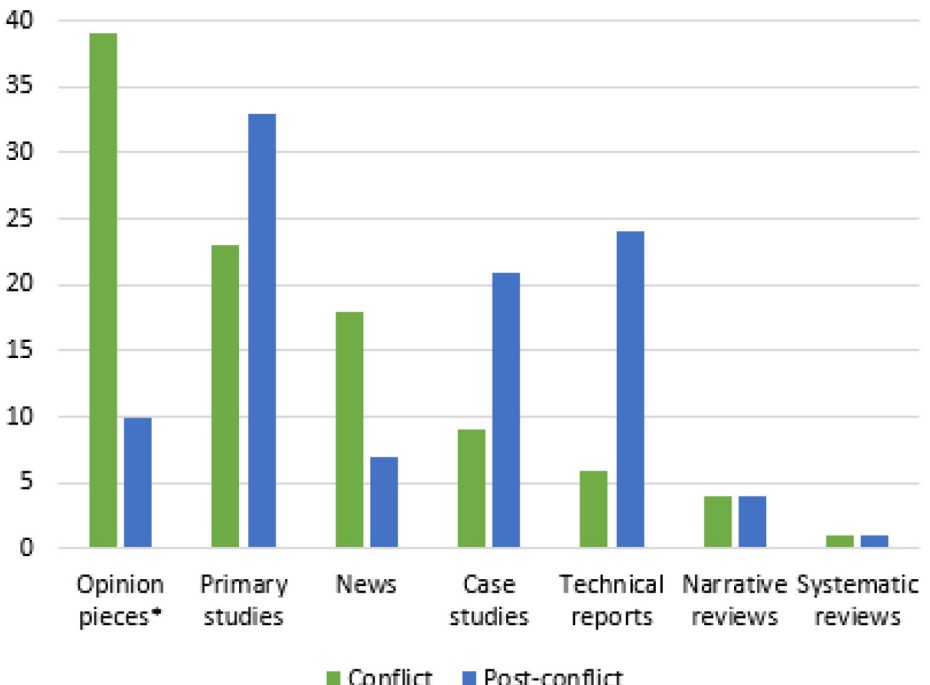

**Fig 4. Types of publication** of the included papers on health care workers in conflict (N = 325) and post-conflict (N = 170) settings.**

**Table 2. Characteristics of authors of the included papers in conflict (N = 325) and post-conflict (N = 170) settings.**

| Any/all authors | Conflict (N = 325) | Post-conflict (N = 170) |
|---|---|---|
| | n (%) | |
| Papers with named authors | 302 (93) | 161 (95) |
| Papers reporting affiliations of authors | 279 (86) | 141 (83) |
| | N = 224* (69) | N = 128* (75) |
| Papers with at least 1 author affiliated with country focus of paper | 90 (40) | 68 (53) |
| % authors affiliated with country focus of paper (median [IQR]) | 0 (0–75) | 20 (0–67) |
| **First author** | **N = 224** | **N = 128** |
| Country focus of paper | 68 (30) | 42 (33) |
| Different country | 151 (68) | 84 (66) |
| - United States of America | 63 (42) | 2 (32) |
| - United Kingdom | 31 (21) | 31 (37) |
| - European countries other than UK | 20 (13) | 7 (8) |
| - Canada | 15 (10) | 7 (8) |
| - Other | 22 (14) | 12 (15) |
| Independent | 5 (2) | 2 (1) |
| **Corresponding author** | **N = 213§** | **N = 125§§** |
| Country focus of paper | 60 (28) | 29 (23) |
| Different country | 148 (70) | 94 (75) |
| - United States of America | 58 (39) | 29 (31) |
| - United Kingdom | 35 (24) | 38 (40) |
| - European countries other than UK | 20 (14) | 8 (9) |
| - Canada | 14 (9) | 7 (7) |
| - Other | 21 (14) | 12 (13) |
| Independent | 5 (2) | 2 (2) |

Abbreviation: IQR, interquartile range

* This is the number of papers reporting affiliations of authors and addressing a specific setting.

§ The corresponding author was unclear in 11 papers.

§§ The corresponding author was unclear in 3 papers.

**Table 3. Funding, conflicts of interest and ethics reporting characteristics of the included papers in conflict (N = 325) and post-conflict (N = 170) settings.**

| | Conflict (N = 325) | Post-conflict (N = 170) |
|---|---|---|
| **Funding sources** | n (%) | |
| Not reported | 263 (81) | 90 (53) |
| Reported as funded | 50 (15) | 75 (44) |
| Reported as not funded | 12 (4) | 5 (3) |
| **Conflicts of interest** | | |
| Not reported | 238 (73) | 106 (62) |
| Reported | 87 (27) | 64 (38) |
| **Ethical approval of primary studies** | **Conflict (N = 76)** | **Post-conflict (N = 56)** |
| Not reported | 33 (44) | 21 (38) |
| Reported as approved | 42 (55) | 33 (59) |
| Reported as not required | 1 (1) | 2 (3) |

(experimental, survey, qualitative, mixed-methods, and document analysis). The maps contain links that redirect the user to PubMed or other databases, to access the title and abstract of included papers, when available.

The systematic map for conflict settings shows that papers on violence and attacks against HCWs were mainly not country specific, about Syria, or about Iraq. The theme of health or medical practice of HCWs was mainly addressed in Iraq and Syria. Education and training of health care workers was the theme mainly addressed in Iraq and Myanmar. Primary studies on HCWs in conflict setting were mainly about Israel and the State of Palestine with a focus on mental health in both countries.

The systematic map for post-conflict settings shows that the theme of workforce performance was mainly about Sierra Leone and Uganda. The papers on workforce supply were mainly not country specific, about Afghanistan, or about Sierra Leone. Primary studies on HCWs in post-conflict setting were mainly about Sierra Leone and Afghanistan with a focus on workforce performance in both countries.

## Discussion

This review presents a systematic mapping of the evidence on health care workers in conflict and post-conflict settings. It has uncovered interesting findings relating to the characteristics of the included papers, journals, and authors respectively; as well as the reporting of funding, conflicts of interest, and ethics.

The systematic map for conflict settings shows the scarcity of primary studies conducted in conflict settings with the predominance of news, opinion pieces and commentaries. This is in line with a previous scoping review on health care workers in the setting of Arab Spring that showed the scarcity of research evidence [7]. Similarly, Patel et al. (2017) reported on the lack of baseline and routine data mainly on violence against health workers [304]. In contrast to conflict settings, primary studies represented the most frequent type of publications on health care workers in post-conflict settings. These findings might relate to the specific settings in which the conflict and post-conflict studies were conducted. However, they might also reflect the challenges in conducting primary research in conflict settings, including security concerns, difficulties in obtaining representative samples and with data collection, political bias, lack of tools and methods specific to conflict settings, and insufficient research funding and capacity [1, 304, 471].

The majority of authors (including first and corresponding) of the included papers were affiliated with high income counties, as opposed to being affiliated with the country focus of the paper. This finding may reflect global imbalances in research capacity between high- and low- and middle-income countries [472–474]. Reasons for this imbalance include limited funding, instability, poor research training, collaboration challenges, and shortage of skilled human resources in low and middle income countries [1, 473–477].

As most of the articles addressing health care workers in conflict and post-conflict are about low-income countries, we can infer that conflicts are still taking place in these countries that already suffer from weak health systems [478]. Our findings are consistent with the previously published scoping review focusing on health care workers in the setting of Arab Spring which found that violence was the most tackled theme [7]. Findings for post-conflict settings concur with a previous review by Roome et al. (2014) on human resources management in post-conflict health systems [5]. This shows the need for more studies on the topic of workforce distribution, which is important to ensure equity in health service provision.

Another interesting finding is the low rates of reporting of funding sources and disclosures of conflict of interest by authors of the included studies. This is particularly for papers about

HCWs in conflict settings. Indeed, funders may have specific agendas while researchers may have political biases and tendency to take sides [471]. These may lead to distorted research and biased data that could be used to mislead local and international communities and negatively affect policy making. Reporting of funding sources and conflict of interest becomes important to better assess the confidence in the publication, particularly when it reports primary studies or makes policy recommendations.

We also found a relatively low reporting of ethical approvals for primary studies, in both conflict and post-conflict settings. This might be attributed to a weak local research capacity including the absence of or complicated ethical review boards [1]. Reporting and seeking ethical approvals in these settings is important given the vulnerability of individuals living in conflict-affected states [479]. This calls journals publishing research conducted in conflict settings to have stringent policies for reporting funding, conflict of interest and ethical approval.

To our knowledge, this is the first systematic mapping of evidence on health care workers in conflict and post-conflict settings. One strength of this study is that we have followed a standardized methodology for conducting and reporting systematic mapping [9]. Further, we have used published frameworks to classify studies on HCWs in conflict and post-conflict settings [5, 7]. One limitation of this study is restricting inclusion to studies published after the year 2000. However, studies published before 2000 might not reflect the current challenges facing health systems and the new aspects of contemporary conflicts. Another limitation of our systematic map is that we relied on the authors' characterization of the conflict (e.g., conflict or post-conflict), and subsequently we did not differentiate between countries in conflict such as Somalia, Iraq and Syria, and those affected by conflict such as Lebanon.

The findings of this review and the resulting systematic maps can support policy makers working on rebuilding health systems post-conflict. These systematic maps provide a comprehensive resource of evidence about HCWs in conflict and post-conflict settings on a global scale. As such, policymakers as well as researchers can use them to find relevant studies by theme. In addition, the mapped evidence can inform policies and practices to protect, support and address the needs of the health care workers in conflict settings. The evidence identified can also inform efforts and strategies for reconstruction and rebuilding of post-conflict health systems, in particular human resource for health.

The findings also highlight the need to strengthen the capacity of local researchers working in conflict-affected states. Also, they can inform agendas of funders and researchers working in the field of health care workers in conflict and post-conflict settings of potential knowledge gaps. This systematic map will inform areas for potential systematic reviews in the field, and may provide a jumpstart for those reviews, given that the relevant studies have already been identified and organized by theme.

## Supporting information

**S1 File. Search strategy.**
(DOCX)

**S2 File. Definition of themes.**
(DOCX)

**S3 File. Geographical map of the included papers.**
(PDF)

**S4 File. Description of the 474 included studies on health care workers in conflict and post-conflict settings (country, setting, study design, and themes).**
(XLSX)

## Acknowledgments

We would like to thank Mr. Mahmoud Chmeiss for designing the geographical map of the included papers, Ms. Nour Hemadi for helping in the screening process and Ms. Karen Bou-Karroum, Ms. Rand Al Ghoussaini and Mr. Mark Jreij for helping in the data abstraction process.

## Author Contributions

**Conceptualization:** Lama Bou-Karroum, Fadi El-Jardali, Elie A. Akl.

**Data curation:** Lama Bou-Karroum, Amena El-Harakeh, Inas Kassamany, Hussein Ismail, Nour El Arnaout, Rana Charide, Farah Madi, Sarah Jamali.

**Formal analysis:** Lama Bou-Karroum, Amena El-Harakeh, Elie A. Akl.

**Methodology:** Lama Bou-Karroum, Tim Martineau, Elie A. Akl.

**Supervision:** Lama Bou-Karroum.

**Validation:** Lama Bou-Karroum, Amena El-Harakeh, Elie A. Akl.

**Writing – original draft:** Lama Bou-Karroum, Amena El-Harakeh, Elie A. Akl.

**Writing – review & editing:** Lama Bou-Karroum, Amena El-Harakeh, Inas Kassamany, Hussein Ismail, Nour El Arnaout, Rana Charide, Farah Madi, Sarah Jamali, Tim Martineau, Fadi El-Jardali, Elie A. Akl.

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
