## [Decision Letter · Decision Letter 0]

6 Jan 2020

PONE-D-19-29429

Health Care Workers in Conflict and Post-Conflict Settings: Systematic Mapping of the Evidence

PLOS ONE

Dear Dr. Akl,

Thank you for submitting your manuscript to PLOS ONE. After careful consideration, we feel that it has merit but does not fully meet PLOS ONE’s publication criteria as it currently stands. Therefore, we invite you to submit a revised version of the manuscript that addresses the points raised during the review process.

We would appreciate receiving your revised manuscript by Feb 13 2020 11:59PM. To enhance the reproducibility of your results, we recommend that if applicable you deposit your laboratory protocols in protocols.io, where a protocol can be assigned its own identifier (DOI) such that it can be cited independently in the future. For instructions see: http://journals.plos.org/plosone/s/submission-guidelines#loc-laboratory-protocols

We look forward to receiving your revised manuscript.

Kind regards,

Jai K Das

Academic Editor

PLOS ONE

Journal Requirements:

2. Please ensure you have carried out your latest search within the last 12 months, or justified your time-frame appropriately.

Reviewers' comments:

Reviewer's Responses to Questions

**Comments to the Author**

1. Is the manuscript technically sound, and do the data support the conclusions?

Reviewer #1: Partly

Reviewer #2: Yes

2. Has the statistical analysis been performed appropriately and rigorously? 

Reviewer #1: N/A

Reviewer #2: N/A

3. Have the authors made all data underlying the findings in their manuscript fully available?

Reviewer #1: Yes

Reviewer #2: Yes

4. Is the manuscript presented in an intelligible fashion and written in standard English?

Reviewer #1: Yes

Reviewer #2: Yes

5. Review Comments to the Author

Reviewer #1: Thank you for sharing this very interesting article on health care workers in conflict and post-conflict settings. While reading the article a few questions arose that I would like the authors to clarify further:

1) I would like some further information on how countries were selected as conflict or post-conflict. I assume it was a dynamic list that categorized countries differently depending on the year? How did the definition of post-conflict took into account the high rates of recidivism where many countries move from conflict to post-conflict to conflict? Similarly after how many years post-conflict was a country no longer defined as post-conflict?

2) On page 7 (and S2) it lists the themes that were identified in the studies, on page 10 the re categorization of the other theme is listed. Please describe this process in text. If papers already had one theme from the original list emerge, were they still additionally coded as other? If not, once those new categories were created did authors revisit old papers to see if they also covered any of these new themes?

3) I am intrigued by the range of study designs that were included, could the authors please add further information about why the decision was made to include grey literature such as opinion pieces.

4) There was limited attention paid to the income level of the country which seemed like an oversight to me given how closely this is related to health workforce and the difference that this may make particularly in the post-conflict period.

As another minor note, I would recommend that the authors read the paper for grammar edits.

Reviewer #2: This is a Systematic Mapping of health sector in conflict and post conflict settings. The paper presents the 'metadata' of such publications and does not focus on the content related issues of conflict and post conflict settings beyond mentioning the themes identified. This should be clarified and explicitly stated.

The paper does not make a distinction between conflict countries [Somalia, Afghanistan] and those affected by conflict [Israel, Lebanon], which is an important one. The authors should consider doing this.

The paper is repetitive at times. It can be made shorter. Detailed comments are provided inside the paper.

I would much prefer to receive a Word version instead of an Adobe version.

6. PLOS authors have the option to publish the peer review history of their article (what does this mean?). If published, this will include your full peer review and any attached files.

Reviewer #1: No

Reviewer #2: Yes: Sameen Siddiqi

---

## [Author Response · Author response to Decision Letter 0]

23 Feb 2020

23 February 2020

Professor Jai K Das

Re: “Health Care Workers in Conflict and Post-Conflict Settings: Systematic Mapping of the Evidence” 

Dear Dr. Das,

Thank you for the opportunity to revise and resubmit our above-titled manuscript to PLOS ONE. We sincerely thank you and the reviewers for taking the time to review the manuscript. We found the comments and suggestions very constructive and used them to improve it.

Kindly find below a point-by-point response to the comments and a description of the changes made to the manuscript.

We look forward to the outcome of the peer review of our revised manuscript.

Sincerely,

Elie A. Akl, MD, MPH, PhD

AUBMC, Department of Internal Medicine

P.O. Box: 11-0236

Riad-El-Solh Beirut 1107 2020

Beirut – Lebanon

Phone: 00961 1 374374

ea32@aub.edu.lb

The editor’s and reviewers’ comments are listed below in bold. Our responses to the comments are in regular font. Relevant extracts from the text are in italic font and changes are underlined.

http://www.journals.plos.org/plosone/s/file?id=wjVg/PLOSOne_formatting_sample_main_body.pdf

http://www.journals.plos.org/plosone/s/file?id=ba62/PLOSOne_formatting_sample_title_authors_affiliations.pdf

Response 1: We revised the manuscript and made all necessary amendments by following the style requirements of PLOS ONE. 

2. Please ensure you have carried out your latest search within the last 12 months, or justified your time-frame appropriately.

Response 2: In the design and conduct of this study, we have adapted to the health field an established methodology for systematic mapping in environmental sciences . Based on this methodology and in reference to the guidance of the Journal of Environmental Evidence and the Collaboration for Environmental Evidence , no more than two years have passed between the searches and submission. We have also consulted with an expert in the systematic mapping methodology to verify this. In addition, this systematic map was conducted in 2017 to gather global evidence on health care workers in conflict and post-conflict settings, after we have conducted a scoping review on health care workers in Syria and Arab Spring (cited in the manuscript) based on a request from the Lancet-AUB Commission on Syria in 2017 .

If the Editors feel strongly about the update of the search, we would be happy to do it, but we will need to ask for an extension.

Reviewer 1:

Thank you for sharing this very interesting article on health care workers in conflict and post-conflict settings.

Response: We thank the Reviewer for this positive feedback.

Comment 1: I would like some further information on how countries were selected as conflict or post-conflict. I assume it was a dynamic list that categorized countries differently depending on the year? How did the definition of post-conflict take into account the high rates of recidivism where many countries move from conflict to post-conflict to conflict? Similarly, after how many years post-conflict was a country no longer defined as post-conflict?

Response: Thank you for this very important insight. Taking into account that countries could transition from one setting (i.e., conflict, post-conflict) to another, we have systematically selected and included papers based on what the papers reported in terms of context (conflict or post-conflict). We tried to avoid making any assumption regarding categorization, given the high rates of recidivism. We have updated the manuscript to clearly explain this point, as per the below (Methods section, Eligibility criteria, pg. 5):

“Setting of interest: We also included both conflict and post-conflict settings. We considered both conflicts between and within states [10]. We focused on contemporary conflicts that started after or were ongoing in the 1990s. Post-conflict settings are considered as a stage of recovery of the state from conflict or crisis and a stage of rebuilding and reconstruction starting from emergency and stabilization followed by transition and recovery, and peace and development [5, 11, 12]. We referred to the description used by the authors when specifying whether the setting of the study was conflict or post-conflict.”

Comment 2: On page 7 (and S2) it lists the themes that were identified in the studies, on page 10 the re categorization of the other theme is listed. Please describe this process in text. If papers already had one theme from the original list emerge, were they still additionally coded as other? If not, once those new categories were created did authors revisit old papers to see if they also covered any of these new themes?

Response: Thank you for your feedback on identifying the themes. The original list of themes was used during abstraction to code data from the included papers (e.g., violence against health care workers, education, etc.). A paper was coded as ‘Other’ in two cases: (1) if it only covered an emerging theme (e.g., medical ethics) or (2) if it covered an emerging theme in addition to an existing one (e.g., education and medical ethics). Indeed, data from one paper can be coded to more than one theme. After finalizing abstraction, we iteratively revisited all papers coded as ‘Other’, to collate data and generate themes. 

To clarify the process, a description has been added to the manuscript as follows (Methods section, Data extraction and coding, pg. 8):

“For the themes, data was coded as ‘other’ if it did not address any of the existing themes, or if it covered an emerging theme and an existing one. Using an iterative process of review and refinement, data coded as ‘other’ was revisited, collated and new themes were generated.

Comment 3: I am intrigued by the range of study designs that were included, could the authors please add further information about why the decision was made to include grey literature such as opinion pieces.

Response: Thank you for your enquiry. In addition to including the peer-reviewed literature, we searched the grey literature, specifically the ReBUILD Consortium Resources webpage and the Human Resources for Health (HRH) Global Resource Center. This is important when addressing the topic of health workers in conflict and post-conflict settings given the scarcity of other types of research papers. For instance, most of the included studies for conflict settings were opinion pieces (39%). It also helps in highlighting where the gap in the literature exists. Grey literature can also provide important contextual information on complex issues .

Comment 4: There was limited attention paid to the income level of the country which seemed like an oversight to me given how closely this is related to health workforce and the difference that this may make particularly in the post-conflict period.

Response: Thank you. The income level of the country is indeed an important factor to be considered for the topic of health workers in conflict and post-conflict settings. As per the reviewer’s suggestion, we have added to the geographical map (Figure 2) a description of the income levels of the countries and described it in the results and discussion sections as follows:

Results section (pg. 9): “The majority of the articles included in this systematic mapping were about low-income countries.”

Discussion section (pg. 17): “As most of the articles addressing health care workers in conflict and post-conflict are about low-income countries, we can infer that conflicts are still taking place in these countries that already suffer from weak health care systems [22].”

Comment 5: As another minor note, I would recommend that the authors read the paper for grammar edits.

Response: We thank the Reviewer for this constructive comment. We have proofread the paper and made all necessary grammatical edits.

Reviewer 2:

Comment 1:

This is a Systematic Mapping of health sector in conflict and post conflict settings. The paper presents the 'metadata' of such publications and does not focus on the content related issues of conflict and post conflict settings beyond mentioning the themes identified. This should be clarified and explicitly stated.

Response: Thank you for emphasizing the need to clarify this in the paper. We have indeed mapped the evidence and used the systematic mapping methodology in this paper to describe the literature across a broad subject of interest, which is in this case, healthcare workers in conflict and post-conflict settings on a global scale. We explicitly mention this in the study objectives in the abstract and introduction (pgs. 2 and 5) as follows: 

“Our ultimate aim is to inform researchers and funders on research gap on this subject and support relevant stakeholders by providing them with a comprehensive resource of evidence about HCWs in conflict and post-conflict settings on a global scale.”

Besides, we have clarified this in the discussion section: This systematic map will inform areas for potential systematic reviews in the field, and maybe provide a jumpstart for those reviews, given the relevant studies have been already identified and organized by theme.

We have also added the below to the methods section (pg. 5) to clarify the scope of our systematic mapping:

“Contrary to systematic reviews, systematic mapping does not aim to answer a specific question but instead “collates, describes and catalogues available evidence (e.g. primary, secondary, quantitative or qualitative) relating to a topic of interest” [9]. In accordance with the definition of systematic maps, this study is a “systematic visual presentation of the availability of relevant evidence” but not the content of the evidence [10] for the topic of healthcare workers in conflict and post-conflict settings. The studies included in a systematic map can be used to identify evidence for policy-relevant questions, knowledge gaps to direct future primary research, and knowledge clusters.

Comment 2:

The paper does not make a distinction between conflict countries [Somalia, Afghanistan] and those affected by conflict [Israel, Lebanon], which is an important one. The authors should consider doing this.

Response: Thank you for your insights. We referred to the description used by the authors when specifying whether the setting of the study was conflict or post-conflict. To account for this, we added the following to the limitations section (pgs. 19-20):

“Another limitation of our systematic map is that we relied on the authors’ characterization of the conflict (e.g., conflict or post conflict), and subsequently we did not differentiate between countries in conflict such as Somali, Iraq and Syria, and those affected by conflict such as Lebanon.”

Comment 3:

The paper is repetitive at times. It can be made shorter. Detailed comments are provided inside the paper.

Response: We thank the reviewer for the suggested modifications and detailed comments in the paper. We addressed all comments and made sure that unnecessary details are removed.

Comments in the PDF:

1. Comments on the abstract:

- “The aim of this study was to systematically identify and map the published evidence on HCWs in conflict and post-conflict settings” (pg. 2)

In order to do what?

What purpose will this mapping serve?

Please elaborate on the objective

Response: We thank the Reviewer for the questions. We have added an explanation to clarify the purpose of this systematic mapping and show its significance. With this comprehensive database of evidence about HCWs in conflict and post-conflict settings globally, the identified evidence can inform efforts and strategies for rebuilding of health systems post-conflict. This has been reflected and emphasized in the study objectives (pgs. 2 and 5) as follows:

“Our ultimate aim is to inform researchers and funders on research gap on this subject and support relevant stakeholders by providing them with a comprehensive resource of evidence about HCWs in conflict and post-conflict settings on a global scale.”

- “For conflict settings, the most studied counties were Iraq (15%), Syria (15%), Israel (10%), and Palestine (9%).” (pg. 2)

Please put the official name of Palestine?

What is the definition of a conflict/post conflict country used in this study? And does Israel fall in that category.

Response: Thank you. We have changed ‘Palestine’ to the official name according to the United Nations (UN), which is ‘State of Palestine’. All necessary changes have been made in the text (pg. 2 in the Abstract, pgs. 10 and 17 in the Results section, and Fig 2).

We have systematically selected and included papers based on what the papers reported in terms of context (conflict or post-conflict). We tried to avoid making any assumption regarding categorization, given the high rates of recidivism. We have updated the manuscript to clearly explain this point, as per the below (Methods section, Eligibility criteria, pg. 5):

“Setting of interest: We also included both conflict and post-conflict settings. We considered both conflicts between and within states [10]. We focused on contemporary conflicts that started after or were ongoing in the 1990s. Post-conflict settings are considered as a stage of recovery of the state from conflict or crisis and a stage of rebuilding and reconstruction starting from emergency and stabilization followed by transition and recovery, and peace and development [5, 11, 12]. We referred to the description used by the authors when specifying whether the setting of the study was conflict or post-conflict.”

As for the definitions, we defined post-conflict settings in the ‘Setting of interest” section (pg. 5) as “a stage of recovery of the state from conflict or crisis and a stage of rebuilding and reconstruction starting from emergency and stabilization followed by transition and recovery, and peace and development.”

For conflict settings, we have added the below definition to the manuscript:

“We focused on contemporary conflicts that started after or were ongoing in the 1990s. We defined conflict as international armed conflicts between two or more states or non-international armed conflicts between non-governmental armed groups or with governmental forces [11]”

As mentioned earlier, we have reflected on the absence of distinction between countries affected by and in conflict in the limitations sections as follows (pg. 19):

“Another limitation of our systematic map is that we relied on the authors’ characterization of the conflict (e.g., conflict or post conflict), and subsequently we did not differentiate between countries in conflict such as Somali, Iraq and Syria, and those affected by conflict such as Lebanon.”

- “countries different from the country subject of the paper.” (pg. 2)

Rephrase. This sentence is not clear

Response: Thank you. For clarification, this has been changed to “countries different from the country focus of the paper”. This has been modified in the whole manuscript.

- “This systematic mapping provides a comprehensive database of evidence about HCWs in conflict and post-conflict settings on a global scale. It can inform policymakers, funders and researchers working in the field of health care workers in conflict and post-conflict settings.” (pg. 2)

For what purpose?

Planning for workforce, reducing emigration, redeployment, effective management etc.

¬Response: Thank you for the helpful suggestions. We have integrated them in the conclusion in the abstract (pg.3) as follows:

“This systematic mapping provides a comprehensive database of evidence about HCWs in conflict and post-conflict settings on a global scale that is needed to inform policies and strategies on effective workforce planning and management. It can also be used to identify evidence for policy-relevant questions, knowledge gaps to direct future primary research, and knowledge clusters. It can inform policymakers, funders and researchers working in the field of health care workers in conflict and post-conflict settings.”

2. Comments on the introduction section:

- “being threatened, detained, and killed” (pg. 3)

Or emigrate. Which is most common in conflict settings

Response: Many thanks, we have added your suggestion to the text as follows:

“HCWs in conflict areas around the world either emigrate due to war or are being threatened, detained, and killed.”

- “While that review revealed a number of themes of interest (e.g., violence against health care workers, education, practicing in conflict setting, migration), it focused on only one region and did not address post-conflict settings.” (pg. 4)

...........other post conflict settings.

Response: Thank you for your comment. The scoping review that we are referring to (Bou-Karroum et al., 2019) did not address any post-conflict setting; it only focused on conflict settings. The other review that we have referenced and referred to looking up for themes on post-conflict setting is a review by Roome et al. on human resources management in post-conflict health systems. 

3. Comments on the methods section:

- “The studies included in a systematic map can be used to identify evidence for policy relevant questions, knowledge gaps to direct future primary research, and knowledge clusters.” (pg. 4)

Please refer to this in the abstract.

Response: Thank you for your insight. We have added this to the abstract as per your suggestion (pg. 3):

“This systematic mapping provides a comprehensive database of evidence about HCWs in conflict and post-conflict settings on a global scale that is often needed to inform policies and strategies on effective workforce planning and management and in reducing emigration. It can be used to identify evidence for policy-relevant questions, knowledge gaps to direct future primary research, and knowledge clusters.”

- “Setting of interest: We considered both conflicts between and within states [10]. We focused on contemporary conflicts that started after or were ongoing in the 1990s. (pg. 5)

Please mention how conflicts were defined

Response: Thank you. We have systematically selected and included papers based on what the papers reported in terms of context (conflict or post-conflict). We tried to avoid making any assumption regarding categorization, given the high rates of recidivism. We have updated the manuscript to clearly explain this point, as per the below (Methods section, Eligibility criteria, pg. 5):

“Setting of interest: We also included both conflict and post-conflict settings. We considered both conflicts between and within states [11]. We focused on contemporary conflicts that started after or were ongoing in the 1990s. We defined conflict as international armed conflicts between two or more states or non-international armed conflicts between non-governmental armed groups or with governmental forces [11]. Post-conflict settings are considered as a stage of recovery of the state from conflict or crisis and a stage of rebuilding and reconstruction starting from emergency and stabilization followed by transition and recovery, and peace and development [5, 11, 12]. We referred to the description used by the authors when specifying whether the setting of the study was conflict or post-conflict.”

- “Study design: We included all types of study designs, including news, editorials/ commentaries/ opinion pieces, technical reports, primary studies, narrative reviews, and systematic reviews” (pg. 5)

Is the Study design not Systematic mapping. Please provide a good reference to this.

https://www.sciencedirect.com/science/article/abs/pii/S138650561830252

Response: Many thanks for sharing this reference. We would be grateful if the Reviewer can reshare the shared link as it did not work. Indeed, our study design is systematic mapping. However, in this section, and as per the systematic mapping methodology, we described the designs of the studies included in our systematic mapping, to show which study designs were eligible for inclusion. This classification allows for mapping the evidence and identifying where gaps in the literature exists. We have previously used this classification in another scoping review on health care workers in the setting of Arab Spring (check reference #7). We provided a reference (reference #9) for the systematic mapping study design as indicated below.

9. James KL, Randall NP, Haddaway NR. A methodology for systematic mapping in environmental sciences. Environmental Evidence. 2016;5(1):7. doi: 10.1186/s13750-016-0059-6.

- “We restricted our eligibility criteria to papers published after the year 2000 to better reflect the current challenges facing health systems and the new aspects of contemporary conflicts.” (pg. 5)

Up to which year?

Response: The search was from 2000 up until 2017. We have clarified this as follows:

“We restricted our eligibility criteria to papers published after the year 2000 up until July 2017 (search date) to better reflect the current challenges facing health systems and the new aspects of contemporary conflicts.”

- We searched the following electronic databases: Medline (Ovid), PubMed, EMBASE (Ovid), the Cochrane Central Register of Controlled Trials (CENTRAL), Cumulative Index of Nursing and Allied Health Literature CINAHL (EBSCOT) on July 2017. (pg. 5)

Was grey literature consulted or only peer review published literature. Such as donor reports, which are frequent and useful source of info in conflict settings.

Response: Thank you for your question. Grey literature has been consulted through searching the ReBUILD Consortium Resources webpage and the Human Resources for Health (HRH) Global Resource Center.

- Data extraction and coding section: The reviewers resolved any disagreement by discussion and when needed with the help of a third reviewer. (pg. 6)

Repetition. Has been stated above.

Response: Thank you. Although we repeated in this section that a third reviewer assists in resolving disagreements, this is specific to the extraction of data from the included studies. It was mentioned in the previous section as a strategy to revolve disagreements while screening studies for potential eligibility. Those two stages are different, and both should be conducted in duplicate and independently. As authors of systematic mapping, we should report on the details of each stage to be transparent about the conduct of the mapping.

4. Comments on the results section:

- Study selection paragraph (pg. 8)

Is this a Result or Methodology?

Response: This section is part of the Results because it includes reporting of the “number of studies screened, assessed for eligibility, and included in the review,” (PRISMA ref) and not how the screening will be done and by whom (which is part of the methodology). This structure is consistent with the published guidance for reporting of systematic maps (James et al., 2016).

- “Fig 1 summarizes the study selection process (S1 Fig).” (pg. 8)

The Fig is much clearer than the narrative. Either explain clearly how you reached the no of 474 from 13863 or just refer to the Figure.

Response: Thank you for your suggestion. We have added the below to the text to refer to the figure for details explaining how the number of screened articles was reduced from 13,863 to 474.

“Please refer to S1 Fig for further details on study selection.”

- “We excluded 968 papers for the following reasons: not study design of interest (n=63); not setting of interest (n=324); not population of interest (n=538);” (pg. 8)

What does this mean - study design not of interest or setting not of interest? Was the study confined to a specific geography? It seemed earlier you were more inclusive than exclusive? 

Response: Thank you for your question. As you mentioned, we were inclusive and did not confine the study to a specific geography or location. All eligible countries have been included in our systematic maps. As per our eligibility criteria reported on page 5, “not study design of interest” means that we only excluded conference abstracts. “Not population of interest” refers to excluding military healthcare workers because we aimed to focus on the delivery of health care primarily to civilians. All of these details are provided in the eligibility criteria in the Methods section.

- Table 1: Countries subject of the included papers in conflict (N=325) and post-conflict (N=170) settings (pg. 9)

Can this not be presented on a global map. Will be more illustrative of the geography of conflicts and post conflict.

Response: Thank you for this great suggestion. We have developed a geographical map to illustrate the geography of conflicts and post-conflicts (S2 Fig and S3 Appendix).

- In Table 1: Countries subject of papers* (pg. 9)

Should the asterisk not be against Not specific?

Response: In Table 1, which shows the countries focus of the included papers in conflict and post-conflict settings, not specific means that the paper is about any conflict or post-conflict setting and does not describe conflict in a specific country. The asterisk is used to explain that a paper can address one or more country, and this is why the percentages do not add up to 100.

- “Most of the included papers on conflict setting addressed the theme of violence against health workers (41%), followed by health or medical practice (34%) and education (21%). For post-conflict settings, besides the themes adopted from Roome et al. (2014) [5], an emerging theme was the mental health of HCWs. The majority of the included papers on post-conflict setting addressed the workforce performance theme (77%) followed by workforce supply (58%).” (pg. 10)

This info is rather limited and cursory. It seems the authors presenting a more in depth description of these themes in another paper?

Response: We have indeed mapped the evidence and used the systematic mapping methodology in this paper to describe the literature across a broad subject of interest, which is in this case, healthcare workers in conflict and post-conflict settings on a global scale. We do not aim to present a more in-depth description of the identified themes in another paper but hope that other researchers and funders will use the generated maps and identified gaps in the literature to conduct future studies.

“Our ultimate aim is to inform researchers and funders on research gap on this subject and support relevant stakeholders by providing them with a comprehensive resource of evidence about HCWs in conflict and post-conflict settings on a global scale.”

We have also added the below to the conclusion in the abstract section (pg. 3):

“This systematic mapping provides a comprehensive database of evidence about HCWs in conflict and post-conflict settings on a global scale that is needed to inform policies and strategies on effective workforce planning and management. It can also be used to identify evidence for policy-relevant questions, knowledge gaps to direct future primary research, and knowledge clusters. It can inform policymakers, funders and researchers working in the field of health care workers in conflict and post-conflict settings.”

And as we have mentioned in the discussion section: This systematic map will inform areas for potential systematic reviews in the field, and maybe provide a jumpstart for those reviews, given the relevant studies have been already identified and organized by theme.

We have also added the below to the methods section (pg. 5) to clarify the scope of our systematic mapping:

“Contrary to systematic reviews, systematic mapping does not aim to answer a specific question but instead “collates, describes and catalogues available evidence (e.g. primary, secondary, quantitative or qualitative) relating to a topic of interest” [9]. In accordance with the definition of systematic maps, this study is a “systematic visual presentation of the availability of relevant evidence” but not the content of the evidence [10] for the topic of healthcare workers in conflict and post-conflict settings. The studies included in a systematic map can be used to identify evidence for policy-relevant questions, knowledge gaps to direct future primary research, and knowledge clusters.

- In Table 3: Country subject of paper (pg. 12)

What does 'country subject' mean? You mean - national of the country.

Response: We have changed ‘country subject of paper’ to ‘country focus of the paper’ to avoid any confusions.

- “The median percentage of authors” (pg. 13)

What is median percentage? Median and IQ range are expressed in percentiles. Please check with a statistician.

Response: Thank you for raising this question. The percentage in this case is not used to summarize a categorical variable across the included papers. It is in fact, a variable at the level of the paper (i.e., every paper has a percentage of authors of affiliated with country subject of paper), and given this is a continuous variables (papers can have a % that spans all the values between 0% and 100%), it is summarized using mean and confidence interval, or using median and IQR (the latter applied in this case given the distribution was not normal).

- In the table Somali-land has been shown as a separate country. Please check if this is recognized by UN. (pg. 14)

Response: Thank you for bringing this up. We have checked the UN classification of countries and Somaliland it considered internationally as part of Somalia. We have included all data on Somaliland under Somalia.

- “The theme of health or medical practice of HCWs was mainly addressed in Iraq and Syria.” (pg. 14)

While the paper touches on the themes but falls short of raising theme related issues.

Response: Thank you for this comment. Although we did not address theme-related issues, we have conducted this synthesis as a systematic mapping to chart the evidence based on a rigorous methodology of collating, categorizing and representing the data. This synthesis provides a comprehensive database of evidence about HCWs in conflict and post-conflict settings that is often needed in planning for workforce, enhancing education, reducing emigration, effective management and retention etc.

5. Comments on the discussion section:

- “The majority of included papers on conflict setting addressed the theme of violence against health workers.

“In the post-conflict setting, the most addressed themes were workforce performance followed by workforce supply.” (pg. 16)

A lot of it is repetition and stated above.

Response: Thank you. We rephrased the statements to avoid repetition (please see below) (pg. 17). In fact, we revised the whole paper and removed any repeated details.

“The majority of included papers on conflict setting addressed the theme of violence against health workers. They mainly related to the conflicts in Iraq and Syria. Similarly, 

Our findings are consistent with the previously published scoping review focusing on health care workers in the setting of Arab Spring found that violence was the most tackled themes [7].”

“In the post-conflict setting, the most addressed themes were workforce performance followed by workforce supply. These Findings for post-conflict settings concur with a previous review by Roome et al. on human resources management in post-conflict health systems [5].

- “One limitation of this study is restricting inclusion to studies published after the year 2000.” (pg. 17)

Another limitation is that the paper does not make a distinction between countries in conflict [e.g. Somalia, Yemen, Syria, Iraq] and those affected by conflict [Lebanon, Jordan, Pakistan].

Response: Thank you for bringing this to our attention. We added it to the limitations as follows (pg. 18):

“Another limitation of our systematic map is that we relied on the authors’ characterization of the conflict (e.g., conflict or post conflict), and subsequently we did not differentiate between countries in conflict such as Somali, Iraq and Syria, and those affected by conflict such as Lebanon.”

---

## [Editor Report · Decision Letter 1]

13 Mar 2020

PONE-D-19-29429R1

Health Care Workers in Conflict and Post-Conflict Settings: Systematic Mapping of the Evidence

PLOS ONE

Dear Dr. Akl,

Thank you for submitting your manuscript to PLOS ONE. After careful consideration, we feel that it has merit but does not fully meet PLOS ONE’s publication criteria as it currently stands. Therefore, we invite you to submit a revised version of the manuscript that addresses the points raised during the review process.

We would like to thank the authors for doing a good job and responding to the peer-review comments adequately.

However for the paper to be considered for publication, there are few following areas for the authors to work on.

- The major findings (HCW characteristics and themes) are not mentioned in the abstract. Please specify these findings in the abstract.

- The primary studies included should be referenced in the results section.

- The authors should add a table describing the characteristics of the included studies as an annex (study ID, types of studies, conflict, HCW involvement, any outcomes, any other relevant information)

We would appreciate receiving your revised manuscript by Apr 27 2020 11:59PM. To enhance the reproducibility of your results, we recommend that if applicable you deposit your laboratory protocols in protocols.io, where a protocol can be assigned its own identifier (DOI) such that it can be cited independently in the future. For instructions see: http://journals.plos.org/plosone/s/submission-guidelines#loc-laboratory-protocols

We look forward to receiving your revised manuscript.

Kind regards,

Jai K Das

Academic Editor

PLOS ONE

---

## [Author Response · Author response to Decision Letter 1]

27 Apr 2020

Professor Jai K Das

Re: “Health Care Workers in Conflict and Post-Conflict Settings: Systematic Mapping of the Evidence” 

Dear Dr. Das,

Thank you for the opportunity to revise and resubmit our above-titled manuscript to PLOS ONE. We sincerely thank you and the Reviewers for the constructive review of our manuscript. Kindly find below a point-by-point response to the comments and a description of the changes made to the manuscript.

We look forward to the outcome of the peer review process.

Sincerely,

Elie A. Akl, MD, MPH, PhD

AUBMC, Department of Internal Medicine

P.O. Box: 11-0236

Riad-El-Solh Beirut 1107 2020

Beirut – Lebanon

Phone: 00961 1 374374

ea32@aub.edu.lb

The editor’s comments are listed below in bold. Our responses to the comments are in regular font. Relevant extracts from the text are in italic font and changes are underlined.

We would like to thank the authors for doing a good job and responding to the peer-review comments adequately. However, for the paper to be considered for publication, there are few following areas for the authors to work on.

The major findings (HCW characteristics and themes) are not mentioned in the abstract. Please specify these findings in the abstract.

Response: We thank the Editor for bringing this to our attention. We have added the following to the abstract to highlight major findings about HVWs characteristics and themes:

Violence against health workers was the most tackled themes of papers reporting on conflict settings while workforce performance was mostly addressed by papers on post-conflict settings. The majority of papers in both conflict and post conflict settings did not report funding sources (81% and 53%), conflicts of interest of authors (73% and 62%) and around half of primary studies did not report on ethical approvals (45% and 41%).

The primary studies included should be referenced in the results section.

Response: Thank you. All necessary changes have been made in the text to include referencing of the included primary studies in the results section (p. 10 of the results section). The included studies can also be found on the interactive gap maps provided in this manuscript. 

The authors should add a table describing the characteristics of the included studies as an annex (study ID, types of studies, conflict, HCW involvement, any outcomes, any other relevant information)

Response: Thank you for this helpful suggestion. We have added a table as an annex (S4 Appendix) to describe the characteristics of the included studies. We refer to it in the text as follows:

Fig 2 presents a geographical map of the countries focus of the included papers (S2 Fig; S3 Appendix). A description of the included studies is also presented in S4 Appendix.

---

## [Editor Report · Decision Letter 2]

13 May 2020

Health Care Workers in Conflict and Post-Conflict Settings: Systematic Mapping of the Evidence

PONE-D-19-29429R2

Dear Dr. Akl,

We are pleased to inform you that your manuscript has been judged scientifically suitable for publication and will be formally accepted for publication once it complies with all outstanding technical requirements.

With kind regards,

Jai K Das

Academic Editor

PLOS ONE
---

## [Editor Report · Acceptance letter]

18 May 2020

PONE-D-19-29429R2 

Health care workers in conflict and post-conflict settings: systematic mapping of the evidence 

Dear Dr. Akl:

I am pleased to inform you that your manuscript has been deemed suitable for publication in PLOS ONE. Congratulations! Your manuscript is now with our production department. 

With kind regards,

on behalf of

Dr. Jai K Das 

Academic Editor

PLOS ONE